

# The Meson Production Targets in the
# high energy beamline of HIPA at PSI

**Daniela Kiselev⋆, Pierre-André Duperrex, Sven Jollet, Stefan Joray,
Daniel Laube, Davide Reggiani, Raffaello Sobbia and Vadim Talanov**

Paul Scherrer Institut, 5232 Villigen PSI, Switzerland

⋆ Daniela.Kiselev@psi.ch

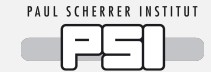

## Abstract

**Two target stations in the 590 MeV proton beamline of the High Intensity Proton Accelerator (HIPA) at the Paul Scherrer Institut (PSI) produce pions and muons for seven secondary beamlines, leading to several experimental stations. The two target stations are 18 m apart. Target M is a graphite target with an effective thickness of 5 mm, Target E is a graphite wheel with a thickness of 40 mm or 60 mm. Due to the spreading of the beam in the thick target, a high power collimator system is needed to shape the beam for further transport. The beam is then transported to either the SINQ target, a neutron spallation source, or stopped in the beam dump, where about 450 kW beam power is dissipated. Targets, collimators and beam dumps are described.**

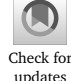
## 3.1 Introduction

The High Intensity Proton Accelerator (HIPA) at PSI [1] delivers a 590 MeV continuous proton beam of up to 2.4 mA, which is accelerated in three stages and described in [2] (of this proceedings). After the Ring cyclotron, the proton beam is sent to the two meson production target stations, M and E [1] [3]. As the Meson Production Target stations have to provide good transmission to the SINQ spallation target, losses due to multiple scattering and nuclear reactions in the targets have to be kept low while keeping the pion/muon yield high. A low $Z$ material is the best target choice according to [4]. In the 1980's, beryllium was used, which failed after a short time at above 120 $\mu$A due to bending stresses on the location of a crack [5]. Another reason for abandon this material was the poisonous and radioactive Be contamination of the surrounding vacuum chamber walls. Since the 1990's graphite has been used for both targets. They last for several years or up to about 40 Ah of proton beam. With a 40-mm (60-mm)-thick target E, the beam transmission is about 70%, (60%). About 10% of the

---

[1]The naming for the two targets, M and E, is derived from the French for thin (mince) and thick (épaisse).

beam is scattered out of the target. For further transport the beam is shaped by a collimator system, where a large fraction of the beam is stopped. Targets, collimators, beam dumps and their environment have to be cooled to dissipate the heat produced. Due to nuclear reactions this area is highly radioactive and needs to be well shielded. Therefore special measures for maintenance have to be considered and provided.

## 3.2 Meson Production Target Stations

Pions are produced by nuclear reactions of the 590 MeV protons with the nucleons in the target above a threshold of about 280 MeV in the center-of-mass frame. Muons are produced by pion decay. When a pion is stopped within 1 mm from the surface of the target, positive muons can escape. These are called surface muons and are used for particle as well as solid-state physics experiments, e.g. the examination of the magnetic properties of materials. Surface muons have energies below 4.1 MeV (corresponding to 29.8 MeV/c) and are almost 100% polarized. Pions exiting the target can produce muons by decay in flight with much higher energies. These are called cloud muons and can have positive or negative charge, although the negative charge is suppressed by a factor 3-4.

Target M feeds two beamlines in the forward direction called PiM1 and PiM3. Target E provides secondary particles for five beamlines, two in forward direction, PiE1 and MuE1, two perpendicular to the proton beamline, MuE4 and PiE3, and one (PiE5) at a backward angle. Muon and pion rates are given in Section 2 [2]. Each target is a 40-cm-diameter graphite wheel that rotates at 1 Hz to distribute the heat spot from the pencil beam. Standard pyrolytic graphite failed due to thermal stress as the expansion coefficients differ strongly in the axial and lateral directions. Radiation induced swelling might have also played a role. Thus polycrystalline graphite from SGL Carbon company is used. It consists of small single crystallites of 10 to 20 $\mu$m, which are irregularly arranged in space. Therefore, the physical properties are almost isotropic, as small grain sizes further improve the isotropy.

### 3.2.1 Target station E

20 (30) kW/mA of power is deposited by the beam in the 40 mm (60 mm) thick target E. At an operating temperature of about 1700 K at 2 mA, the target is cooled primarily by radiation due to the large emissivity of graphite. Water-cooled copper shields are mounted on the rear of the target within the vacuum chamber to dissipate the heat. As the target is mainly surface cooled, the maximum temperature is approximately independent of the target thickness. However, since the beam losses are higher with the 60 mm target, the maximum beam current for a 60-mm thick target is limited to 2 mA due to cooling issues.

The target with its shielding plug (Figure 3.1 right) is inserted vertically into the beamline. As a consequence, the horizontal rotating shaft has to be small and so the two bearings must be close to the target. For this reason, heat transfer to the bearings has to be reduced by proper target design. For this, the graphite and the hub with the bearings are connected by only six hollow spokes, which maintain the target shape but can also follow dimensional changes due to thermal expansion. After 2002, the graphite rim has been separated into 12 segments by slits of 1 mm to reduce deformation of the rim (Figure 3.2 left). Before, the radial deformation of the graphite wheel was observed to increase with rising beam current. This could cause the proton beam to partly miss the target as its width is just 6 mm. The small width favours surface muons from the produced pion distribution, which roughly follows the beam shape. It also keeps the temperature gradient between the center and the surface of the target small, which reduces thermal stress. However, it requires that the proton beam is always well centered. This is accomplished by a beam centering system relying on the beam position monitors in front of the target stations. Further, the transmission of the beam is controlled constantly and

a deviation leads to a beam interlock as a pencil like beam missing the target could damage the SINQ target.

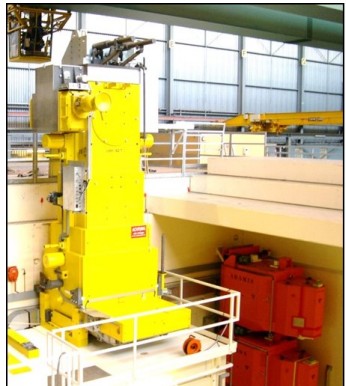
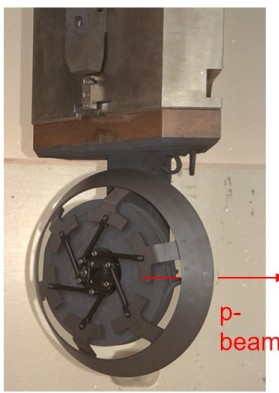

Figure 3.1: Left: Exchange flask (yellow) for the Target E insert. In the background in orange is the exchange flask of Target M. Right: Target E insert with the old graphite wheel design.

Recently, the sensitivity to deviations of the beam from the center of the target was significantly improved with a modified version of the graphite wheel. For this, small grooves on both sides of the graphite target were applied (see Figure 3.2 right). In this way, the beam transmitted through the target is modulated, when it deviates more than 0.5 mm from the target centre. From there on, the amplitude of the modulation depends strongly on the position of the beam. Since different spacings are used between the grooves inside and outside, a deviation left and right from the center can be identified. More details, including information about the Fast Fourier Transformation used for the signal analysis, can be found in [6].

As the bearings degrade from heat and radiation, they have to be replaced after a few months of operation. First, several meters of concrete have to be removed from beamline. Then the target insert with the shielding plug is pulled into the exchange flask by remote control. The 42-t exchange flask (Figure 3.1 left) is well shielded by up to 40 cm steel for the up to 3 Sv/h graphite wheel [7]. The same shielding flask is used for removing collimators and beam dumps out of the beamline. The exchange flask is transported with the 60 t crane to a door lock above the service cell (ATEC) at PSI. The door lock is remotely opened by the control unit of the exchange flask. Then the target insert is lowered into the service cell,

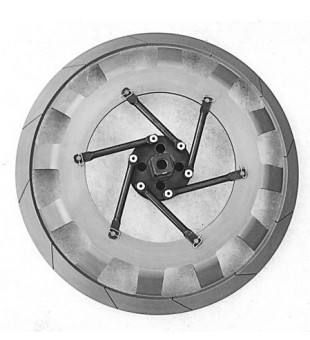
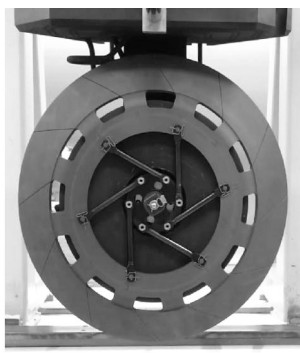
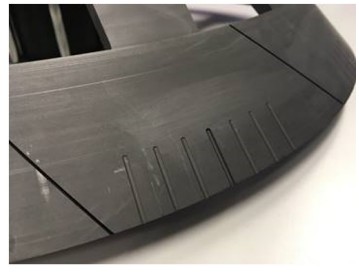

Figure 3.2: Different Target E types from 2002 on. Left: Graphite wheel with 12 segments. Middle: Slanted target type. Right: Target E with grooves.

which is equipped with manipulators for remote handling. The hub with the two bearings are exchanged using these manipulators. During scheduled user beam time, the second target insert, which is fully equipped and has been stored in a vacuum chamber, is put back into the beamline to reduce the downtime.

A new type of target wheel was successfully tested at the end of 2019. Unlike the standard wheel, the beam here passes with a small angle through the graphite, keeping the effective target thickness (40 mm) the same (see Figure 3.2 middle). This configuration, called slanted target, results in a larger active surface and has two locations, the entrance and exit of the beam, where the beam is close to the surface. Both effects lead to an increase of surface muons. A first analysis [8] indicates an increase of 40 - 50%.

### 3.2.2 Target station M

As the Target M has a much smaller thickness, and the bearings are far from the beam and placed in the shielding, the demands are much less challenging than for Target E. The rim of the target is about 2-cm wide with a thickness of 2 mm. As the beam passes through the rim at an angle of $30^o$, its effective thickness is 5.2 mm (see Figure 3.3 left). This leads to a beam loss of only about 1.6% in the target and the following collimator system. The power deposition is about 2.4 kW/mA and the target operates at around 1100 K, mainly cooled by thermal conduction.

The original design dates back to 1985. The 85-cm steel shielding plug is placed upstream of the target and is not accessible during beam operation. The target insert is mounted horizontally, which has the advantage that the rotating shaft is long and the two bearings are well shielded. This results in bearing lifetimes of several years. In 2012/13 a new target insert was designed and installed in the beamline (see Figure 3.3 right). The bearing lifetime is improved due to better cooling of the front of the shielding plug close to target and beam. Here an additional copper plate cooled by water, is attached. The rotating shaft is made of low conducting material, titanium-vanadium, to reduce the heat flux from the target to the bearings. In this design the bearings can be exchanged without changing the target by pulling the shaft through the shielding plug. Further improvements in the maintenance and handling of the vacuum seal at the rear of the target insert were implemented in the new design.

In the near future, precision particle physics experiments will require higher rates, particularly for surface muons, to stay at the forefront of muon intensity. HIMB, High Intensity Muon Beam, aims to increase the surface muon rate with a 20-mm thick slanted target design and beamlines transporting a large fraction of the secondary particles produced. An increase of two orders of magnitude in the rates for surface muons is envisaged.

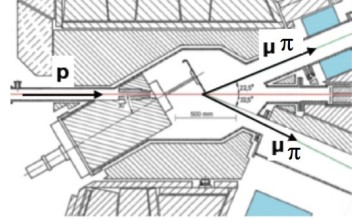
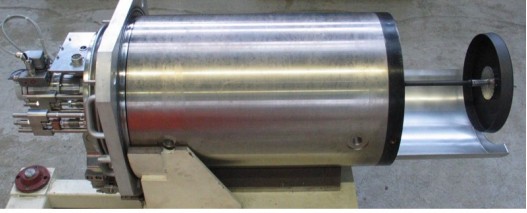

Figure 3.3: Left: Schematic view of the Target M insert at the beamline. Right: The Target M insert, new design.

## 3.3 High Energy Collimators and Beam dump

As the collimators and beam dumps have to stand high power from the proton beam, both devices are similar in their design. Like Target E, they are inserted vertically and contain steel shielding above the component. Collimators and beam dumps are both made from oxygen-free, high purity copper for three reasons: to have good thermal conductivity, to avoid hydrogen embrittlement and for brazing of the steel tubes onto the copper body. Hydrogen embrittlement occurs at high temperatures and can lead to cracks. The hydrogen is not an impurity in the copper but produced by spallation reactions of the protons with copper. Hydrogen bonds to the oxygen present in copper as impurity to form water, which then causes cracks at elevated temperatures. Brazing requires an oxygen-free surface. However, during brazing at temperatures around $800^{o}$C oxygen diffuses out of the copper and passivates the surface leading to a bad junction and thermal contact.

Cooling is quite important to avoid not only melting but temperatures above the homologous temperature (half of the melting temperature in Kelvin), where the structure of the material starts to change significantly. Therefore, temperatures above $400^{o}$C in copper must be avoided. Since direct contact of the water with the proton beam is not recommended due to the production of aggressive ions that lead to corrosion, the water pipes are wound outside of the cylindrical body. With a water flux of about 8 m/s the tubes cannot be made from copper, since they would suffer from abrasion, which leads to erosion corrosion. Therefore steel tubes are brazed to the copper body, which requires a good thermal contact in between. Before a new device is put into the beamline, the thermal contact is tested by heat exchange experiments.

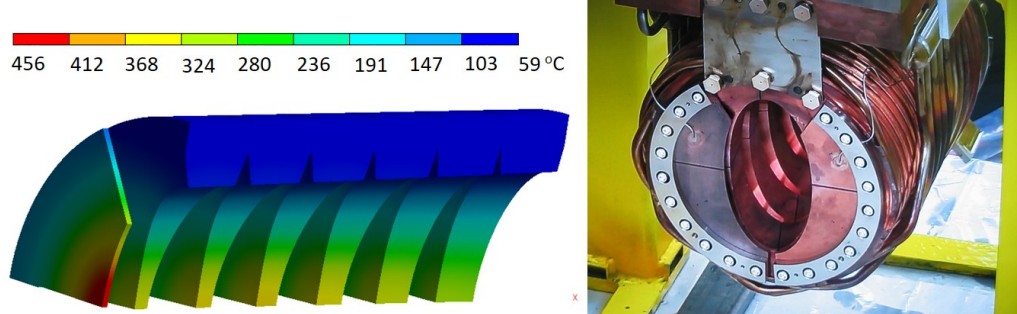

Figure 3.4: Left: Temperature distribution of the KHE2 with 2 mA beam. Right: Collimator KHE2 with sample plate from the backside.

The cylindrical copper body is composed of five or six slices, which are later brazed together. This shape cannot be manufactured from one block, since some of the slices are tailored on both sides to reduce the energy deposit of the proton beam by reducing the amount of material. Slits between the slices also help to reduce the thermal stress. Each slice also contains four radial slits for thermal expansion. The optimal shape of the collimator has to be found by computational fluid dynamics (CFD) or equivalent simulations, which take into account the actual distribution of the proton beam or import the energy deposition region-wise from particle transport Monte Carlo simulations. The temperature distribution inside the collimator KHE2 along the beam direction is shown in Figure 3.4 left. More than one device is often necessary to absorb the beam under the constraint that the maximum temperature is kept below the homologous temperature and that the device still fits in the exchange flask. In fact, the collimator system after Target E and the beam dump each consist of four parts. The maximum length of each part is 400 mm. The collimator system after Target E is distributed along 4 m, whereas the beam dump sections are separated from each other by about 100 mm.

An aperture, separated in four sectors, is mounted in front of most devices. It consists of 100 $\mu$m Nickel foil, where free electrons from ionization due to protons are collected. This signal is proportional to the fraction of the beam in a section, and serves as an indication of the beam position as well as the beam size. The aperture is used to protect the device behind with a machine interlock, if the beam properties deviate from normal.

The KHE2, the third collimator after Target E, absorbs between 100 kW and 140 kW of the beam depending on the beam tuning and the thickness of Target E. This means that a large fraction of the beam hits the collimator and might cause radiation damage. An early estimate using the particle transport Monte Carlo package MCNPX2.5.0 [9] predicted an average DPA (Displacements Per Atom) of around 20. Regions close to the beam have an up to four times higher DPA value. Therefore, visible signs of radiation damage were expected and the collimator was inspected in the hot and service cell ATEC of PSI. The inside of the collimator was examined by an inspection tool to avoid high doses to the camera [10], which was well shielded without direct view to the collimator. This was necessary due to a dose rate of 310 Sv/h, 10 cm from the entrance of the collimator. No cracks or serious damage were observed except for some pieces peeling off the collimator. These pieces were identified as graphite (by the grey color) as well as due to the presence of $^7$Be, a typical radioisotope from carbon activation. The graphite likely sublimated from Target E. A sample was taken and later a measurement with a HPGe (High Purity Germanium) detector was performed. In addition, traces from the brazing material, such as silver isotopes, were found. In 2013 the KHE2 was replaced by a new collimator of identical design, but with more thermocouples and additional sample plates from copper and Glidcop, a copper matrix with 0.3 wt % aluminum oxide, for later material studies after irradiation (See Figure 3.4 right). Glidcop is a promising candidate with similar properties as copper but keeping a large fraction of the thermal conductivity under irradiation.

In the meantime a new collimator system KHE2 and KHE3 with a different inner shape was manufactured, which will stand up to 3 mA beam current. The maximum current for the present KHE2 is 2.15 mA according to CFD simulations, which use the physical and mechanical properties for unirradiated copper. The main difference in the design is that the inner cone of the present collimator KHE2 has a diameter that widens in beam direction, whereas in the new design it decreases as in the beam dumps. Therefore, on the slices in front of the new designed KHE2 much less beam power is absorbed as the cone opening at this position is much wider. A side effect is that the slices are only slightly tailored. With the new design beam transport with a 3% larger transmission is possible up to the SINQ target.

In 2016 a sudden increase of the vacuum pressure inside the beam tube in the vicinity of the beam dump indicated a malfunctioning component. However, it was not clear which component was causing the problem. A mass spectrometer connected to the beam tube indicated the presence of water, which restricted the leak to a component cooled by water. However, there are many components, such as slits, vacuum chambers and beam dumps, connected to the cooling water cycle. The leak appeared at a beam current above 1.4 mA measured in front of the 40-mm Target E. However, it was very difficult to locate the leak, since it could not be detected without beam, and also did not show up when the device was heated with $150^o$C pressurized water. Since the full beam dump consists of four parts, the malfuntioning part had to be identified before a replacement could be manufactured. The leak was identified with beam studies, and finally confirmed when the leak disappeared in 2018 with a new identical BHE1 in place. During the time the first beam dump section was removed from the beamline and transferred to ATEC for inspection and replacement, a periscope using mirrors was inserted into the beamline at the position of BHE1. A camera at its end took pictures from the second part of the beam dump as well as the entry of the vacuum chamber. A view on the BHE2 from this camera is shown in Figure 3.5 left. On the right of the figure, BHE2 is shown

before irradiation. As can be seen from the pictures BHE1 is intact despite withstanding 150 kW with beam.

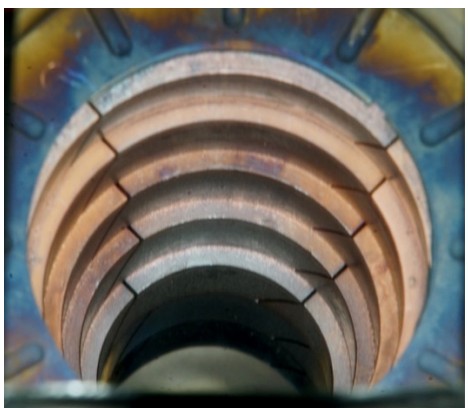
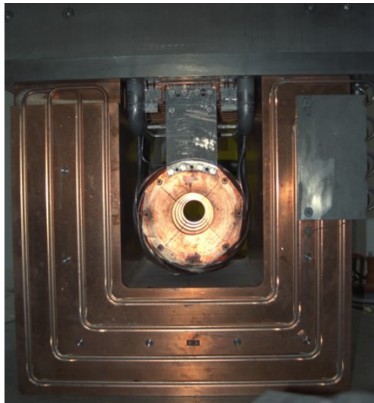

Figure 3.5: Left: Beam dump BHE2 with aperture in the beamline as seen from the periscope. Right: BHE2 without aperture before irradiation in 1990.

### 3.4 Summary

The two meson production stations M and E use rotating polycrystalline graphite targets. They have been working well since the 1990's, serving seven beamlines with pions and muons. A special target design with grooves was recently tested and allows a very precise detection of the beam position on the target. For HIMB aiming to increase the surface muon rate by up to a factor of 100, beamline simulation and design studies for an upgrade of the target M station with the new type of slanted target design are ongoing. In the Target E station the slanted target type already demonstrated a 40-50% increase of the surface muon rate.

The collimator system as well as the beam dump have to stand more then 100 kW per component. Except for a water leak in the first beam dump element, which is likely due to thermal cyclic stress, no visible signs of radiation damage are observed. The design of a segmented copper body cooled by water in steel tubes, which are brazed to the copper, has proven its reliability.

### Acknowledgments

We would like to thank Gerd Heidenreich, the inventor of the target stations, collimators and beamdumps, and Ake Strinning, a versatile engineer, for their ideas and the long lasting design in a harsh radioactive environment. For supporting many ANSYS simulations for the HIPA high energy beam line as well as for his dedicated detailed geometrical models in MCNPX, we thank Michael Wohlmuther, the former group leader of "Radiation transport and multiphysics" at PSI. Pedro Baumann we thank for the excellent maintenance and improvement of the target and beamline components. Further we thank Thomas Rauber for taking care of the secondary beamlines and for the inspection of the beam dump region with the periscope. Finally we thank the many specialized technical groups for supporting the 590 MeV beamline and target region at HIPA.

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
