# Peer review of "The Meson Production Targets in the high energy beamline of HIPA at PSI"

_SciPost Physics Proceedings, doi:SciPost Phys. Proc. 5, 003 (2021)_

## Round 1 · Referee Report · Anonymous (Referee 1) · 2021-6-11

Report
This paper is very interesting, clearly written and deserves publication. I have only very few remarks: 1) In the Abstract, I did not understand the number 450 kW beam power in the beam dump. The total beam power at 2.4 mA at 590 MeV is 1.4 MW. Is 1 MW lost in Target E and the collimators?
2) I think there is a typo in line 206: inspection ans replacement. Should "ans" not be "and"? 3) G. Heidenreich, M. Wohlmuther and U. Rohrer have contributed considerably to the whole project. If not in the list of authors, it might be polite to acknowledge their contributions

---

## Round 1 · Referee Report · Adrian Signer (Referee 2) · 2021-6-17

Report
opportunity to review an earlier draft of the article and were in
communication with the authors before the submission. All our comments
and suggestions have been taken into account. Hence, we think the
paper can now be published in the current form.

---

## Round 1 · Referee Report · Claude Petitjean (Referee 3) · 2021-6-21

Strengths
1) detailed and comprehensive review of PSI's meson production target stations 2) good description of the history and how technical problems were solved and improved 3) nice illustrations of the target wheels and related components
Weaknesses
1) the english could be improved by a professional person with native English origin
Report
Requested changes
1) Abstract, last sentence: leave off "In the following" and"further" 2) lines 15 and 43: say ".. described in Ref. [2] (section 2).." to make clear that it is not a section of this paper 3) line 16: leave off "last stage of the" 4) line 21: it would be appropriate to mention that the Beryllium targets were also abandoned because of the evaporation of the Be above 120 uA causing poisonous and radioactive contamination of all surrounding vacuum chamber walls. 5) line 28: insert "special": therefore special measures .. 6) line 31: leave off "mainly" 7) line 38: leave off "up to 60 MeV" (there is no real limit) 8) line 39: "by a one order of magnitude": it is rather "by a factor 3-4 9) line 206 typo "and replacement" not "ans" 10) references: It would be nice if all references show the year of appearance. in refs 1,3,8 it is missing 11) it would be appropriate to mention somewhere, that the names M&E come from the french words "mince" (thin) and épaisse (thick)! (invention of Prof. H.J. Gerber)

---

## Round 2 · Referee Report · Claude Petitjean (Referee 3) · 2021-7-6

Strengths

1) this paper gives an excellent account of PSI's high energy target stations, collimators and beam dump, its functioning, especially concerning the heat dissipation generated by the MW primary proton beam. 2) the many figures are very informative and make the paper easy to read and understand.

Report

In the second version of this paper all minor critics have been corrected and suggestions included. Thus, it is ready for publication in the SciPost journal.

Requested changes

no requests

---

## Round 2 · Author Response

Dear reviewer, thank you for the review and the kind words about our manuscript. 1) The 450 kW dissipated energy for the beam dump looks indeed surprising and you are not the first one wondering about. First, using a 40 mm target, the beam dump stands only 1.6 mA as maximum current on Target E. In Target E and collimator system about 30 % is lost. Therefore, we end up with 1.15 mA or 644 kW (using 575 MeV as some energy is lost in the Target E). The value of 450 kW is calculated from the water flux and the temperature difference of out-going to in-coming water. This value fits very well to an old calculation of Gerd Heidenreich. Another 50 kW is expected for the local shielding left and right of the collimator and 40 kW distributes in the shielding above the collimator according to simulations. The remaining energy is dissipated in the surrounded shielding. 2) Yes, indeed this is a typo. Thank you. It is corrected in the new version. 3) Gerd Heidenreich is indeed the developer of both target stations, the collimator system and the beam dump. I am already in contact with him and he agreed to review our design for the HIMB station. It is a pleasure for me to acknowledge him for the design, which lead to a long lasting and high performing target station. Urs Rohrer was not involved in the target station development but into the beam line design. However, a large part of the success of the target stations as well as the exchange flasks can be devoted to Ake Strinning, the engineer bringing the design ideas of Gerd to life. Michael Wohlmuther was mainly involved in the simulations for the SINQ and UCN target stations, not in the technical design or operation of the meson target facilities. However, as a group leader he was supporting the ANSYS simulations and I will acknowledge him for this.

With kind regards,

Daniela Kiselev

Dear Claude, thank you for the positive rating of our paper and the in-depth review. All corrections were made as you suggested. Below you find a few comments to the most interesting ones: 4) Be contamination: Although I was expecting this problem, I never explicitly heard about. Probably the so called “Be flushing”, which sucks the air into the vacuum pipe to avoid spreading of the contamination, solved the issue. It is still used during the target exchange. 8) Ratio of positive to negative cloud muons: Indeed, it is more a factor 3-4 than an order of magnitude. I probably mixed it with surface muons. 10) Years of references added. 11) Good idea to mention the reason for the naming. Now I know even who it invented!

Best regards, Daniela

---

## Round 2 · List of Changes

The most important changes are listed above.

---

## Editorial Decision

published